# A Prospective Cohort Study Assessing the Relationship between Plasma Levels of Osimertinib and Treatment Efficacy and Safety

**DOI:** 10.3390/biomedicines11092501

**Published:** 2023-09-10

**Authors:** Tatsuro Fukuhara, Kazuhiro Imai, Taku Nakagawa, Ryotaro Igusa, Hayato Yokota, Kana Watanabe, Aya Suzuki, Mami Morita, Ren Onodera, Akira Inoue, Masatomo Miura, Yoshihiro Minamiya, Makoto Maemondo

**Affiliations:** 1Department of Respiratory Medicine, Miyagi Cancer Center, Natori 981-1239, Japan; fukuhara-tatsuro@miyagi-pho.jp (T.F.); watanabe-ka750@miyagi-pho.jp (K.W.); suzuki-ay806@miyagi-pho.jp (A.S.); mamimorita1980@yahoo.co.jp (M.M.); 2Department of Thoracic Surgery, Akita University Graduate School of Medicine, Akita 010-8543, Japan; i-karo@mui.biglobe.ne.jp (K.I.); minamiya@jd6.so-net.ne.jp (Y.M.); 3Department of Thoracic Surgery, Omagari Kosei Medical Center, Daisen 014-0027, Japan; drtakubo@yahoo.co.jp; 4Department of Respiratory Medicine, Osaki Citizen Hospital, Osaki 989-6136, Japan; ryo-igusa@h-osaki.jp; 5Department of Pharmacy, Akita University Hospital, Akita 010-8543, Japan; hayato@hos.akita-u.ac.jp; 6Division of Pulmonary Medicine, Department of Internal Medicine, Iwate Medical University School of Medicine, Yahaba 028-3694, Japan; aiueonoderaiueo@gmail.com; 7Department of Palliative Medicine, Tohoku University Graduate School of Medicine, Sendai 980-8575, Japan; akira.inoue.b2@tohoku.ac.jp; 8Department of Pharmacokinetics, Akita University Graduate School of Medicine, Akita 010-8543, Japan; m-miura@hos.akita-u.ac.jp; 9Division of Pulmonary Medicine, Department of Medicine, Jichi Medial University, Shimotsuke 329-0498, Japan

**Keywords:** osimertinib, EGFR, trough level, non-small cell lung cancer

## Abstract

Osimertinib is a standard treatment for patients with EGFR-mutated non-small cell lung carcinoma (NSCLC). We evaluated the relationship between plasma osimertinib concentrations and treatment outcome in patients with NSCLC for this cohort study. The plasma levels of osimertinib and its metabolite AZ5104 were measured a week after the start of treatment (P1). The primary endpoint was to evaluate the correlation between plasma concentration and adverse events (AEs). The correlation with treatment efficacy was one of the secondary endpoints. In patients with CNS metastases, the concentration in the cerebrospinal fluid was also measured. Forty-one patients were enrolled. The frequency of AEs was highest for rash, followed by anorexia and thrombocytopenia. Thirty-eight cases provided measurements for P1. The median plasma concentration of osimertinib was 227 ng/mL, and that of AZ5104 was 16.5 ng/mL. The mean CNS penetration rate of two cases was 3.8%. The P1 in the group with anorexia was significantly higher than that in the group without anorexia (385.0 ng/mL vs. 231.5 ng/mL, *p* = 0.009). Divided into quartiles by P1 trough level, Q2 + Q3 (164–338 ng/mL) had longer PFS, while Q1 and Q4 had shorter PFS. An appropriate plasma level of osimertinib may avoid some adverse events and induce long PFS. Further large-scale trials are warranted.

## 1. Introduction

EGFR (epidermal growth factor receptor) mutation is a major oncogenic driver in non-small cell lung carcinoma (NSCLC), and inhibition of EGFR signaling by EGFR-tyrosine kinase inhibitors (TKIs) represent the optimal clinical treatment strategy for NSCLC patients with EGFR mutations. Currently, the approved EGFR-TKIs in Japan are gefitinib and erlotinib as the first generation, afatinib and dacomitinib as the second generation, and osimertinib as the third generation. Osimertinib is an irreversible EGFR inhibitor, in contrast to gefitinib and erlotinib. Osimertinib monotherapy is one of the standard treatments for patients with EGFR-mutated lung cancer. Using osimertinib as the first-line treatment led to a statistically significant difference in PFS and OS when compared to gefitinib or erlotinib [1]. Osimertinib treatment had a median PFS of 18.9 months and a median OS of 38.6 months. In the phase III AURA3 study, osimertinib treatment for tumors with the resistance mutation T790M after failure of the initial treatment with EGFR-TKIs was evaluated. The median PFS and OS for osimertinib treatment were 10.1 and 26.8 months, respectively [2]. Frequent adverse events in AURA3 were diarrhea, rash and paronychia. However, predictors of efficacy and safety of osimertinib treatment have not been elucidated. According to pharmacokinetics data, osimertinib has a mean half-life of 48.3 h, with a peak-to-trough ratio of 1.6 [3]. Ethnicity and food intake did not affect exposure. Osimertinib was metabolized to produce at least two circulating metabolites, AZ5104 and AZ7550, which accounted for less than 10% of all osimertinib-related products [4,5]. In particular, AZ5104 binds more strongly to wild-type EGFR as well as mutated EGFR, and may be involved in adverse events associated with wild-type EGFR inhibition. 

Previous studies have reported a positive association between trough levels of gefitinib, erlotinib or afatinib and several adverse events [6,7]. Conversely, in some reports, the association is negative [8,9]. Similarly, regarding the relationship between trough concentration and PFS, there are positive reports [10,11,12], whereas there are also negative reports [9,13], and no conclusion has been reached regarding the relationship between trough levels and the efficacy and toxicities of first- and second-generation EGFR-TKIs. Regarding osimertinib, there have been several reports on the relationship between the blood levels of osimertinib and toxicity or efficacy. There is a report showing a relationship with toxicity [14], and a report showing that high concentrations may shorten OS [15]. On the other hand, there is also a negative report [16]. 

To assess the usefulness of the measurement of plasma osimertinib or AZ5104 trough concentration, we evaluated the correlation between trough concentration and safety or efficacy.

## 2. Methods

### 2.1. Study Design

This was a prospective observational study of NSCLC patients with an acquired EGFR T790M mutation who received osimertinib treatment. The study was conducted in compliance with the Helsinki Declaration, the Japanese Ethical Guidelines for Medical and Health Research Involving Human Subjects, and the Japanese ethical guidelines for human genome and gene analysis research. Moreover, the study protocol was reviewed and approved at the local IRB existing in participating institutions. A total of 6 institutes and hospitals participated and enrolled patients in this study in Japan (Clinical Trial ID: UMIN000024327).

### 2.2. Patients

Patients with non-small cell lung cancer that was histologically or cytologically confirmed and with stage IIIB, IV, or postoperative recurrence, with an EGFR-activated mutations (exon 19 deletion or exon 21 L858R), who were found to have acquired a T790M-resistance mutation after first- or second-generation EGFR-TKI treatment and were planning to undergo osimertinib treatment as the next regimen were eligible. In addition, other eligibility criteria were at least one measurable lesion on a RECIST (version 1.1) assessment, ECOG performance status 0–2, maintenance of organ function suitable for osimertinib treatment, and written informed consent, along with being over 20 years of age. The main exclusion criteria were: interstitial pneumonia or pulmonary fibrosis evident on chest radiograph, uncontrolled angina, myocardial infarction within 3 months, heart failure, uncontrolled diabetes and hypertension, severe infections, and significant or recent gastrointestinal disorders with diarrhea. 

### 2.3. Procedures

Blood samples were collected in a collection tube (EDTA-2K), centrifuged (2000× *g*/10 min/room temperature) for 1 h, and plasma components were transferred to sterile spitz and cryopreserved at −20 °C. Plasma samples were collected at five time points: first, between enrollment and the start of study treatment (P0); second, at 8 days after the start of the study treatment (P1), the time to reach >95% steady-state trough concentration [17]; third, at the time of adverse events that are expected to require suspension for over one week or discontinuation of the treatment (P2); fourth, at 8 days from the recommencement of study treatment (P3); and fifth, at the end of osimertinib treatment due to tumor progression (P4). The P1 sample was obtained to measure the basic plasma trough concentration. When cerebrospinal fluid was collected from a patient with brain metastases for medical needs, the concentration of cerebrospinal fluid (CSF) was optionally measured. When calculating the CSF penetration rate, since the cerebrospinal fluid collection time and plasma collection time were different, we used an osimertinib population pharmacokinetic model to predict the plasma concentration from the trough level, adjusting to the cerebrospinal fluid collection time [18].

### 2.4. Measuring of Plasma and CSF Concentration

The concentration was measured via the high-performance liquid chromatography (HPLC) method, which was previously described [19]. In brief, a Capcell Pak C18 MG II (250 mm × 4.6 mm ID; Shiseido, Tokyo, Japan) HPLC column was used for drug analysis. The mobile phase was a mixture of 0.5% KH2PO4 (adjusted to pH 3.5), acetonitrile, and methanol (55:25:20). The flow rate was 0.5 mL/min at ambient temperature, and sample detection was carried out at 250 nm. A 10 μL solution of gefitinib (10 ng), as an internal standard for the quantitation of osimertinib and AZ5104, was added to the 100 μL plasma or CSF sample, followed by dilution of the sample with 900 μL of water and vortexing for 30 s. This mixture was applied to an Oasis hydrophilic–lipophilic balance extraction cartridge that had been previously activated with methanol and water (1.0 mL each). The cartridge was then washed with 1.0 mL of water and 1.0 mL of 40% methanol in water followed by elution with 0.4 mL of 100% methanol and then 1.0 mL of 100% acetonitrile. The residue was dissolved in 20 μL of methanol, 20 μL of the mobile phase was added to the sample, and a 20 μL aliquot of the sample was then processed on the HPLC system. 

### 2.5. Outcomes

The primary endpoint was to assess the correlation between plasma levels of osimertinib or AZ5104 and adverse events. There were three secondary endpoints—the first was to evaluate the correlation between plasma concentration of osimertinib or AZ5104 and the therapeutic effect, and the second was the correlation between the CSF concentration of osimertinib or AZ5104 and CNS-related toxicity. The third was to clarify the pharmacological genomics of osimertinib. 

### 2.6. Case Number Setting and Statistical Analysis 

In phase I trials, high doses of 160 mg and 240 mg increased the frequency of diarrhea and rash [20]. Namely, a high plasma concentration was expected to be associated with a higher incidence of diarrhea and rash. According to the Japanese patient subgroup analysis (*n* = 80) of the combined results of the phase II part of the international joint phase I/II study (AURA study) and the phase II study (AURA2 study), the incidence of diarrhea and rash were 36.3% and 56.3%, respectively (data from the package inserted at the time of approval) [21]. If the number of cases is 40, the number of cases that develop diarrhea is calculated to be 15, and the number of cases that develop rash is 23 cases, which is enough to withstand the analysis. As for cerebrospinal fluid, since there are few data in humans, we aimed to analyze 10 cases. 

All *p*-values were two sided, and a *p*-value of 0.05 or less was considered statistically significant. All statistical analyses were performed with EZR (Saitama Medical Center, Jichi Medical University, Saitama, Japan, version 1.61), which is a graphical user interface for R (The R Foundation for Statistical Computing, Vienna, Austria, version 4.2.2). More precisely, it is a modified version of R commander (version 2.8-0) designed to add statistical functions frequently used in biostatistics [22]. Kaplan–Meier survival curves were constructed for progression-free survival and compared using a stratified log-rank test. HR and associated 95% CIs were calculated using Cox proportional hazard analysis. Multivariate Cox regression models were used for the adjusted comparison of progression-free survival between treatment groups. The quartiles of osimertinib trough concentration on day 8 (P1) were defined as follows: the lower quartile (Q1) was the 25% of patients with the lowest values; quartiles 2 and 3 (Q2 and Q3) comprised patients whose trough levels were within the interquartile range; the upper quartile (Q4) was the 25% of patients with the highest values. These three categories (Q1, Q2–Q3, and Q4) were used for stratification as appropriate.

## 3. Results

### 3.1. Patients Characteristics and Clinical Results

Forty-one patients were enrolled in the study (Table 1). The median age was 68 years (range 43–81), with 33 females (80%). There were six ECOG PS 2 patients (14.6%) and two PS 3 patients (4.9%). The number of Stage IV patients was 32 (78%). The previous treatment history was one regimen in 20 patients (48.8%), two regimens in 6 patients (14.6%), and three or more regimens in 15 patients (36.6%). Thirty-four cases (82.9%) had CNS metastases. The overall response rate for osimertinib treatment was 53.7%, and the disease control rate was 92.7% (Table 2). The median PFS was 6.7 months and the median OS (defined as the time from the initiation of osimertinib to death) was 19.9 months (Figure 1). The frequency of adverse events was highest for rash, with that of all grades being 36.6% and that of G3 or higher being 2.4% (Appendix A). This was followed by anorexia (all grades: 31.7%), thrombocytopenia (all grades: 29.3%), anemia (all grades: 29.3%), and diarrhea (all grades: 26.8%). The frequency of drug-induced pneumonitis was 19.5% for all grades and 2.4% for grades of G3 or higher.

### 3.2. Osimertinib and AZ5104 Concentration

Thirty-eight cases provided measurements for the trough concentrations in plasma at 8 days after the start of treatment (P1). The median concentration of osimertinib was 227 ng/mL, and that of AZ5104 was 16.5 ng/mL (Figure 2, left column). No correlation was found between these trough concentrations and body surface area or body mass index (Appendix A). CSF samples were obtained from two patients (Table 3). The osimertinib concentrations were 8.2 ng/mL and 10.1 ng/mL, respectively. On average, the penetration rate was 3.8%.

### 3.3. P1 Trough Level and Adverse Events

The cases in which each adverse event occurred were collected and the median P1 trough concentrations were compared (Figure 2). In comparison with the osimertinib trough concentration, adverse events that were above the median for all cases were neutropenia (*n* = 6, 385 ng/mL), leukocytopenia (*n* = 6, 385 ng/mL), anemia (*n* = 12, 372 ng/mL), anorexia (*n* = 13, 339 ng/mL), nausea (*n* = 7, 339 ng/mL), and rash (*n* = 15, 235 ng/mL). On the other hand, in comparison with AZ5104 concentration, increased concentrations were observed in paronychia (*n* = 10, 19.7 ng/mL), in addition to the neutropenia, leukopenia, anemia, and anorexia observed with osimertinib. Analysis of the P1 trough concentrations of osimertinib and AZ5041, divided into the occurrence group and the non-occurrence group for each adverse event, showed no difference in adverse events, except for anorexia (Figure 3 and Appendix A). Concentrations in the anorexia occurrence group were significantly higher for both osimertinib and AZ5401 than for trough concentrations in the non-occurrence group (osimertinib: 385.0 ng/mL vs. 231.5 ng/mL, *p* = 0.009 and AZ5041: 37.8 ng/mL vs. 17.1 ng/mL, *p* = 0.005). In the analysis of both osimertinib and AZ5401, pneumonitis was not related to the plasma level of both.

### 3.4. P1 Trough Level and Treatment Efficacy

In order to evaluate the correlation between the trough concentration and the therapeutic effect, scatter plots of the trough concentration and PFS of each case are shown in Figure 4A,C. It was suggested that PFS tended to be longer in cases where the trough concentration of osimertinib was within the middle range. Next, the patients were divided into the quartile groups according to the osimertinib trough levels at P1; patients in lowest quartile (Q1) had osimertinib trough levels of less than 164 ng/mL; the intermediate quartile of patients (Q2 and Q3) had osimertinib trough levels ranging from 164 to 227 ng/mL and 227 to 338 ng/mL, respectively; and patients in the highest quartile (Q4) had osimertinib trough levels of over 338 ng/mL. The PFS of Q1, Q2, Q3 and Q4 were compared (Figure 4B), and the median PFS of the Q2 group and Q3 group were 15.3 months [95%CI: 0.5–NA] and 26.5 months [95%CI: 2.8–NA], respectively. The Q2 + Q3 group around the median (164–338 ng/mL) had the longest PFS compared to the Q1 group and the Q4 group (22.8 months [95%CI: 6.7–36.4] vs. 4.6 months [2.5–7.3] vs. 5.1 months [95%CI: 0.5–6.5], respectively). The hazard ratio for Q2 + Q3 was 0.020 [95% CI 0.14–0.84, *p* = 0.020] compared to Q1. Thus, PFS was significantly prolonged in cases with P1 trough concentrations ranging from 164 to 338 ng/mL. When classified into quartile groups according to the AZ5041 trough concentration, the patient group with the lowest trough level (Q1) had a trough level of less than 6.8 ng/mL, the patient group with 25–50% trough level (Q2) had a trough level of 6.8–16.5 ng/mL, and the patient group with 50–75% trough level (Q3) had a trough level of 16.5–29.1 The group with the highest trough level (Q4) had a trough level of 29.1 ng/mL or higher. The Q3 patient group tended to have long PFS (22.8 months, Figure 4D). Additionally, univariate and multivariate analyzes of PFS showed that the osimertinib trough level of the Q2 + Q3 group was significantly associated with longer PFS compared to the Q1 + Q4 group (Table 4). In multivariate analysis, the duration was significantly longer in the never-smoking group compared to the smoking-history group. Next, we investigated the frequency of adverse events in each quartile group classified according to osimertinib trough concentration (Table 5). The Q4 group included more patients who had AEs ≥ G3, and 30% of cases discontinued study treatment due to AEs (Figure 4A,C). 

### 3.5. P2 Trough at the Time of Adverse Events

In seven cases, the trough concentration at the time of AE occurrence (P2) could be measured. The relevant adverse events were gastrointestinal toxicity in four cases and pneumonitis in three cases. There were no differences in blood count, liver function, or renal function, However, the osimertinib and AZ5104 concentration at the time point of P2 were significantly higher than those for P1 (Table 6). Regarding the trough concentration at the time of resumption with a dose reduction to 40 mg after an adverse event (P3), the concentration of osimertinib decreased (range: 143 to 290 ng/mL), indicating that the dose reduction to 40 mg was appropriate (Appendix A). In addition, comparing the final plasma concentrations of eight cases measured at PD (P4) and five cases measured at AE discontinuation (P2), the plasma concentrations of the cases showing long PFS were kept low (range: 112 to 177 ng/mL) (Figure 5).

## 4. Discussion

In this observational study, we evaluated the association between osimertinib treatment and plasma concentration in previously treated EGFR mutation-positive lung cancer. Regarding plasma trough concentration, a statistically significant higher concentration was shown in patients with anorexia, and this high concentration was also observed in cases with gastrointestinal toxicity and hematological toxicity. Although trough levels could be measured in only a few cases upon the occurrence of AEs (P2), the cases tended to show high trough concentrations with the occurrence of AEs. PFS was significantly longer in the Q2 + Q3 group (164–338 ng/mL, 22.8 months) than in the lower trough Q1 (4.6 months) or higher trough Q4 (5.1 months) when grouped by quartile according to osimertinib trough concentration. 

In this study, the median PFS was 6.7 months and median OS was 19.9 months. These were shorter than the study AURA3, which included cases with similar conditions. The reasons for this poor prognosis were that we enrolled many cases of poor performance status, with over three previous regimens, and most of the cases had CNS metastasis, representing common cases in clinical practice.

The penetration rate of osimertinib to CSF was 3.8% of the estimated plasma value. It can be said that this is a high value, even when compared with other EGFR-TKIs. The permeability of gefitinib, erlotinib, and afatinib to CSF was 1.1–1.8%, 2.8%, and 2.5% of plasma trough values, respectively [23,24,25]. As for osimertinib, the median CSF penetrance was reported to be 0.79% in seven patients [26]. Although there is a discrepancy with our study, the higher penetrance rate in our study can explain the actual efficacy of osimertinib, which is said to be effective for CNS metastases. The number of cases is small in both studies, and verification in a larger number of cases is necessary. 

In this study, cases with anorexia had higher trough levels. In addition, the concentration of gastrointestinal toxicity such as nausea and vomiting tended to be high. Other cases of hematological toxicity tended to have high trough concentrations. However, no association with trough concentration was found for diarrhea, rash, pneumonitis and liver dysfunction. Optimal management of trough levels after the start of treatment may reduce the incidence of anorexia. On the other hand, even if the trough concentration 8 days after the start of treatment was appropriate, there were cases in which the estimated trough concentration at the onset of AE was high. It cannot be ruled out that the concentration increases due to the patient’s metabolic capacity and the effects of drug interactions. AZ5104 binds directly to the wild-type EGFR and was speculated to be associated with adverse events, but its concentration is less than one-fifth to one-tenth that of osimertinib, and this study could not confirm its usefulness versus osimertinib.

It was shown that PFS may be prolonged in cases (Q2 + Q3 group) with appropriate trough concentration 8 days after the start of osimertinib treatment. In Q1, the concentration is low and might indicate that the patient received undertreatment with osimertinib. On the other hand, in Q4, AE is likely to occur, and there are many cases of AE discontinuation. 

The limitations of this study were that it was a prospective observational study, and the number of cases was small. In particular, a more large-scale validation of spinal fluid testing and blood testing at the time of AE occurrence is needed. For the examination of AZ5104, the concentration was one-tenth of that of osimertinib, and some cases were close to the limit of detection sensitivity. In the pursuit of even more accurate data, even more sensitive testing methods may be needed. Although it was not possible to analyze the factors that determine trough levels, this study showed that measuring plasma concentration has great clinical importance. Developing factors to predict trough concentration is an important future challenge.

## 5. Conclusions

It was shown that trough concentration measurement 8 days after the start of osimertinib may be able to predict several toxicities and efficacy in EGFR-mutated tumors. An appropriate plasma level of osimertinib may avoid some adverse events and induce long PFS. Further analysis is required.

## Figures and Tables

**Figure 1 biomedicines-11-02501-f001:**
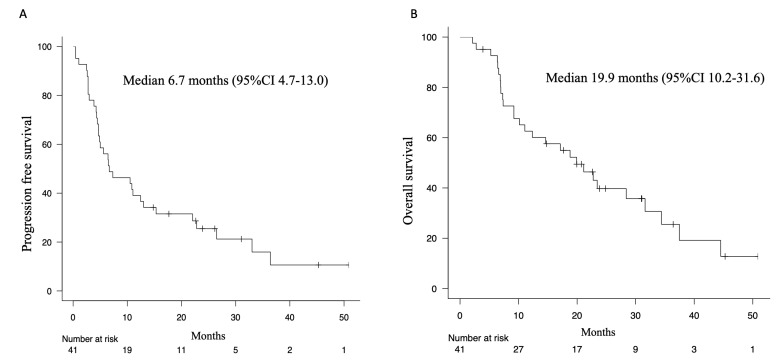
Kaplan-Meier curves for progression free survival (**A**) and for overall survival (**B**) of osimertinib treatment for all enrolled population.

**Figure 2 biomedicines-11-02501-f002:**
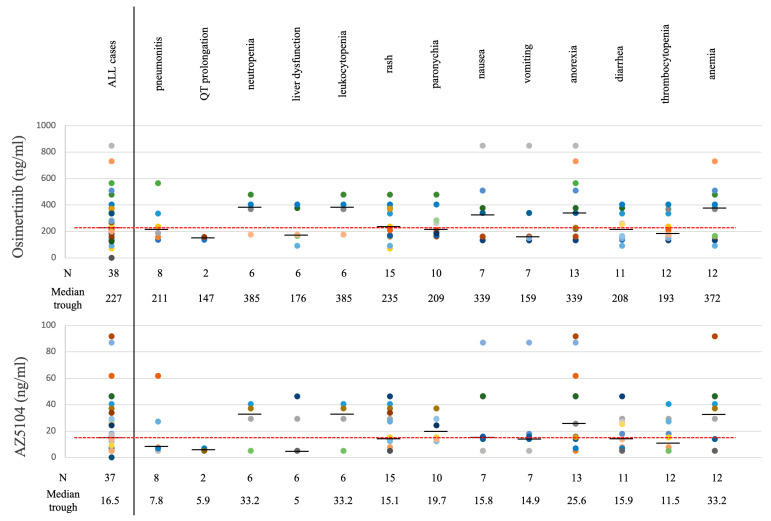
Trough levels eight days after the start of osimertinib treatment (P1) in patients with each adverse event. Number of cases and median trough concentrations. (**Top**) Osimertinib; (**Bottom**) AZ5104. Individual cases are indicated by the same color of the circle.

**Figure 3 biomedicines-11-02501-f003:**
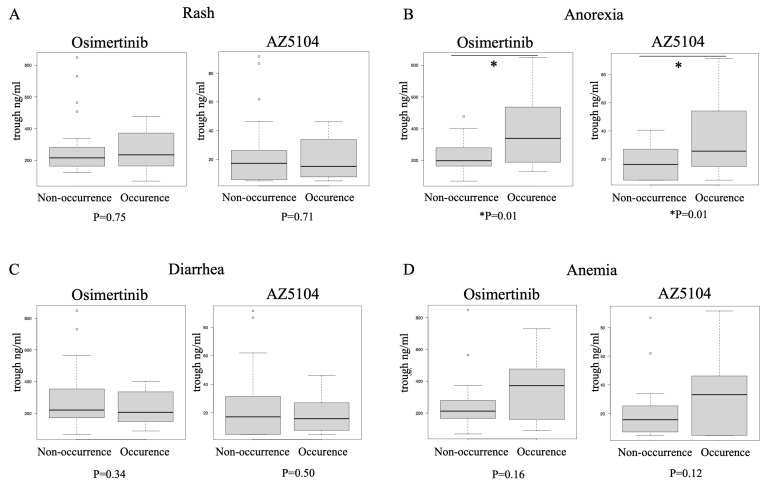
Comparison of trough levels eight days after the start of osimertinib treatment (P1) between the occurrence group and the non-occurrence group of each representative adverse event. (**A**). Rash, (**B**). Anorexia, (**C**). Diarrhea, (**D**). Anemia.

**Figure 4 biomedicines-11-02501-f004:**
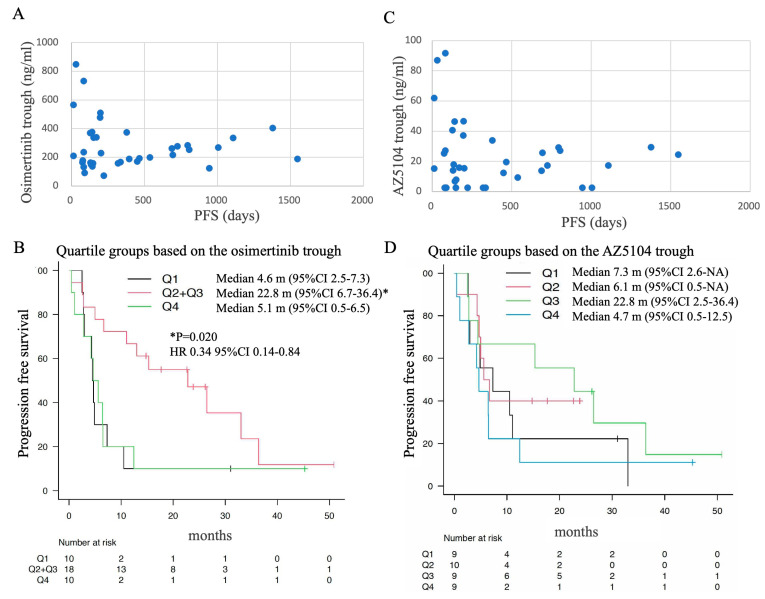
Correlation between osimertinib or AZ5104 trough levels eight days after the start of osimertinib treatment (P1) and treatment efficacy. (**A**,**C**) Scatter plot of PFS and trough level. (**A**) Osimertinib; (**C**) AZ5104. (**B**,**D**) Kaplan-Meier curves for the PFS of patients classified into quartile groups according to the trough levels. (**B**) Osimertinib; (**D**) AZ5104. Patients were divided into quartile groups according to osimertinib trough levels at P1. Q1 has trough levels below 164 ng/mL, Q2 has trough levels between 164 and 227 ng/mL, Q3 has trough levels between 227 and 338 ng/mL, and Q4 has trough levels above 338 ng/mL.

**Figure 5 biomedicines-11-02501-f005:**
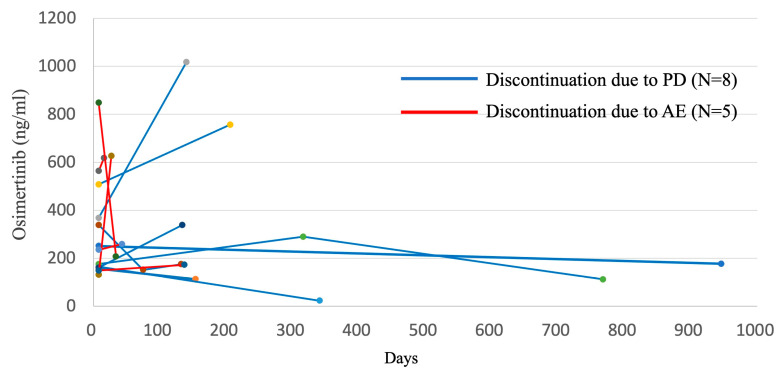
Changes in osimertinib trough concentrations by cause of discontinuation. The cases indicated by blue lines were due to tumor progression, and the cases indicated by red lines were due to adverse events.

**Table 1 biomedicines-11-02501-t001:** Patients’ characteristics.

N = 41	*n*	(%)
Median age	68	43–81 (range)
Sex		
Male	8	(20)
Female	33	(80)
ECOG PS		
0	17	(41.5)
1	16	(39.0)
2	6	(14.6)
3	2	(4.9)
Smoking status		
Former or current smoker	11	(26.8)
Never smoker	30	(73.2)
Stage of disease before first line treatment (7th)		
IV	32	(78.0)
Recurrence	9	(22.0)
Number of previous regimens		
1	20	(48.8)
2	6	(14.6)
≥3 (range 3–8)	15	(36.6)
CNS metastases		
positive	34	(82.9)
negative	7	(17.1)
EGFR mutation subtype		
Exon 19 del	27	(65.9)
Exon 21 L858R	14	(34.1)
Exon 20 T790M	41	(100)

**Table 2 biomedicines-11-02501-t002:** Treatment response.

Response	N	%
CR	0	0%
PR	22	53.7%
SD	15	36.6%
PD	1	2.4%
NE	3	7.3%
Response Rate		53.7%
Disease Control Rate		92.7%

**Table 3 biomedicines-11-02501-t003:** The results of CSF analysis.

Case	Plasma Trough (ng/mL)	CSF Level (ng/mL)	Plasma Level of the Same Time as CSF (ng/mL) *	CSF/Plasma
OC-02	186	10.1	238	4.2%
OC-03	191	8.2	248	3.3%

* Predicted the plasma concentration from the trough level adjusting to the cerebrospinal fluid collection time using the osimertinib population pharmacokinetic model [18].

**Table 4 biomedicines-11-02501-t004:** Univariate and multivariate PFS analyses.

Variables		Univariate Analysis		Multivariate Analysis	
		HR (95%CI)	*p*-Value	HR (95%CI)	*p*-Value
Age	<75 y vs. ≥75 y	1.995 (0.907–4.384)	0.086	2.173 (0.873–5.406)	0.095
Sex	Female vs. Male	1.170 (0.524–2.611)	0.701	2.710 (0.835–8.794)	0.097
Stage	Postoperative recurrence vs. Stage III, IV;	2.983 (1.057–8.418)	0.039	2.199 (0.555–8.719)	0.262
Smoking	Never vs. Former/Current	0.636 (0.283–1.425)	0.271	0.284 (0.089–0.903)	0.034 *
previous regimens	1 vs. >2	1.050 (0.520–2.123)	0.892	1.051 (0.391–2.828)	0.922
EGFR mutation	L858R vs. Ex 19 del	1.106 (0.522–2.347)	0.792	1.159 (0.480–2.800)	0.743
Brain metastasis	No vs. Yes	1.223 (0.588–2.542)	0.590	1.737 (0.590–5.117)	0.317
Osimertinib	Q2 + Q3 vs. Q1 + Q4	2.704 (1.235–5.920)	0.013	4.492 (1.643–12.280)	0.003 *

* *p* < 0.05.

**Table 5 biomedicines-11-02501-t005:** Osimertinib trough concentration and frequency of adverse events.

	P2 Cases	Cases with Grade 3 or Higher AE	Cases Discontinued due to AE	Cases Discontinued due to PD
Q1 (*n* = 10)	3 (30%)	6 (60%)	1 (nausea) (10%)	7 (CNS 2) (70%)
Q2 + Q3 (*n* = 18)	3 (16.7%)	2 (11.1%)	2 (ILD 2) (11.1%)	2 (CNS 1) (11.1%)
Q4 (*n* = 10)	2 (20%)	4 (40%)	3 (nausea *, anorexia **, and ILD) (30%)	5 (CNS 0) (50%)

***** In patients with nausea, the dose was reduced to 40 mg, but the nausea did not improve and treatment was discontinued. ** The other patient with anorexia refused to continue treatment when anorexia occurred.

**Table 6 biomedicines-11-02501-t006:** The comparison of plasma concentration and serum analysis at P1 and P2 (N = 7).

Plasma Concentration		P1		P2		*p*
Osimertinib	ng/mL	216	(90.6–564)	618	(164–832)	0.022 *
AZ5104	ng/mL	17.9	(n.d.–61.9)	33.4	(16.6–92)	0.031 *
Serum		P1		P2		*p*
white blood cell	×10^3^/μL	42	(29–48)	44	(30–61)	0.553
neutrocyte	×10^3^/μL	26	(18–29)	30	(22–50)	0.128
hemoglobin	g/dL	10.5	(8–13.3)	10.4	(7.6–12.6)	0.059
platelet	×10^3^/μL	135	(123–338)	193	(123–266)	0.834
aspartate aminotransferase	U/L	21	(15–45)	17	(12–27)	0.172
alanine aminotransferase	U/L	14	(7–34)	11	(5–25)	0.14
total bilirubin	mg/dL	0.32	(0.15–0.6)	0.33	(0.21–0.66)	0.498
serum creatinin	mg/dL	0.72	(0.58–1.08)	0.77	(0.56–0.99)	0.352

* *p* < 0.05.

## Data Availability

The data presented in this study are openly available after publication.

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
