# Peer review of "A Prospective Cohort Study Assessing the Relationship between Plasma Levels of Osimertinib and Treatment Efficacy and Safety"

_biomedicines, 2023, doi:10.3390/biomedicines11092501_

Round 1
Reviewer 1 Report
pleaase change anorexia-occurred with group with anorexia
Smoking exposure could affect the treatment efficacy and tolerability by activating cytochrome P450 enzymes.
Why was only an exclusion criterion chosen? What about pack-years and comorbidities?
Pleaase explain how were the staging determined? Was a liquid biopsy performed for determining the acquired resistance? How was OS calculated?did it start from the progression after EGFR first-line? Please include the perspectives of this study.
The statistical methods should be better described, was a cox regression analysis performed?
I suggest to include the following references for the discussion about EGFR acquired resistance
-Future Oncol. 2018 Jun;14(13s):29-40
-Future Oncol. 2016 Sep;12(18):2149-61.
Reviewer 2 Report
This is a study of the plasma levels of osimertinib and evaluated them with efficacy and safety.
The authors reported some adverse effects were associated with high plasma levels and the patients with appropriate trough concentration prolonged PFS.
Two patients in Q4 discontinued osimertinib due to AE (nausea or anorexia). Did they try the reduced dose of osimertinib or other EGFR TKI? Usually, such adverse events are manageable and OS may same.
Generally, the plasma levels of osimertinib are not affected by body weight, gastric pH, serum albumin, concomitant drug usage, age, or ethnicity.
Are there any factors associated with the plasma levels of osimertinib? In other words, can we predict high plasma levels, or can we detect who should be started with a low dose of osimertinib?
Is the day 8 the best time? The plasma levels of osimertinib were achieved on day 15. Do you have any preliminary experiments on the timing?
As for CSF exams, when were they collected? Were the timing of the plasma test and CSF test the same?
Reviewer 3 Report
In this work, Fukuhara et al. disclose the correlations between plasma levels and osimertinib efficacy and safety in NSCLC patients. Osimertinib is a promising drug for the treatment of advanced NSCLC, and the submitted manuscript is informative and well-written. Its topic and content suits well to this journal. The presentation of the results needs some revision, as well as some other stylistic matters. Thus, I recommend minor revision:
Abstract: Please mention in the abstract that NSCLC patients were evaluated.
Introduction: Please mention that osimertinib is an irreversible EGFR inhibitor, in contrast to gefitinib and erlotinib.
The tables need some more information and explanation in the headers or in footnotes, e.g., a specification of the used abbreviations.
Figure 1, caption: Specify ´´ITT´´.
Figure 4: Please mention briefly the trough levels of each quartile group in the caption again.
Table 5: Please modify the style of this table according to the journal guidelines.
Table 6: Please correct ´´10^3´´.
References: I suppose the references need to be adjusted to the journal style.
